# Pruning harvesting with modular towed chipper: Little effect of the machine setting and configuration on performance despite strong impact on wood chip quality

Alessandro Suardi[1], Sergio Saia[2]*, Vincenzo Alfano[1], Negar Rezaei[3], Paola Cetera[4], Simone Bergonzoli[1], Luigi Pari[1]

1 Council for Agricultural Research and Economics -Research Centre for Engineering and Agro-Food Processing (CREA-IT), Monterotondo, Roma, Italy, 2 Department of Veterinary Sciences, University of Pisa, Pisa, Italy, 3 National Research Council (CNR) Research Institute on Terrestrial Ecosystems (IRET), Porano, TR, Italy, 4 Dipartimento di Agraria, Università degli Studi di Sassari, Sassari, Italy

* sergio.saia@unipi.it

**Data Availability Statement:** All relevant data are within the paper and its Supporting Information files.

## Abstract

Pruning residues can have a high quality as feedstock for energy purposes and are largely available in Europe. However, it is still an untapped resource. Such scarce use is due to the need to optimize their supply chain in term of collection machines and the associate cost of collection. A modular chipper (prototype PC50) for pruning harvest was developed. Such prototype is adaptable to various harvesting logistics and may produce a higher quality woodchip compared with the one produced by shredders available in the market. In this work, we tested the performance and quality of the product delivered by the prototype PC50 in various conditions and plant species, after a modulation of the machine settings (counter-rotating toothed rollers [CRR] speed), loading systems ([LS], either big bag or container), and knife types ([KT], either discontinuous hoe shaped knives or continuous helicoidal knives). To take into account of the covariates in the experiment (Cropping season and plant species), LSmeans were computed to have an unbiased estimate of the treatments means. The modulation of LS and KT scarcely affected the performance of the machine. In particular, the choice of the KT affected the field efficiency when the LS was a Tilting box but not a Big Bag. Whereas the continuous knife resulted in a 97% higher material capacity compared to hoe shape knives, the last of which the amount of short sized (<16 mm) fractions compared to helicoidal knives. No role of the CCR was found on the machine performance, but increasing CRR speed reduced the chip apparent bulk density and the fraction chips with a size <8 mm.

## 1. Introduction

Bioenergy plays a significant role in climate change mitigation [1] since it contributes to reduce the use of fossil fuels for energy production. The agricultural sector is one of the main

**Funding:** This study has been produced with the data collected during the EuroPruning project (Development and implementation of a new and non-existent logistics chain for biomass from pruning); and the European Horizon 2020 project Magic (grant number 727698). The ideas expressed do not represent either those of the European Commission or the Italian Ministry of Agriculture (MiPAAF).

**Competing interests:** The authors declare that no competing interests exist.

suppliers of biomass through bioenergy crops or using plant and animal residues [2]. In particular, the residues are constituted by the non-edible plant fractions usually not collected and left on the field surface [3, 4], e.g. the straw of cereals or the pruning residues. Considering the European Renewable Energy Directive [5], the advantages of using agricultural residues for energy production are, on the one hand, the reduction of the land dedicated to the cultivation of bioenergy crops avoiding the competition with food crops and on the other hand to turn an untapped resource with a disposal cost into an economic advantage for farmers [6]. Indeed, residuals have a relatively low cost [7–11], and their use can consist in an indirect positive effect for the environment.

Hence, bioenergy is a tool to improve the economic and environmental sustainability of agricultural sector. Pruning residues are a good quality feedstock for energy purposes and are also hugely available [6, 12–14]. In Europe pruning biomass available yearly accounts for more than 25 million tons [15]. In EU-28, the most widespread fruit trees are olive grove (more than 0.5 M ha), apple (>0.5 M ha), almonds (almost 0.8 M ha), peaches (>0.2 M ha 209450 ha), and plums (>0.15 M ha) [16].

Although fruit tree pruning can be an abundant biomass resource, their exploitation for energy purposes is still limited. On the contrary, pruning is usually disposed through open-air burning which is a costly and environmental harmful practice. Otherwise, pruning wastes are mulched and left on the ground and/or incorporated into the soil [17]. However, such a latter use is scarcely applied.

Due to the lack of a well-organized pruning biomass supply chain in Europe, there is no actual market for pruning residual. However, in recent years, the growing demand for biomass for energy use has led equipment manufacturers to focus their attention on the management of pruning and on finding solutions for recovering such a huge amount of biomass [18].

The first attempt consisted of adapting mulchers, which were used to incorporate these wastes into the soil; interest later shifted towards chippers and balers [19], which were able to produce a high-grade fuel. Indeed, comminution and storage processes have a strong influence on important parameters of the biomass, including the presence of contaminants (soil, stones), and its particle size and bulk density, which may impact their performances when used [20].

Hence, the residue harvesting process is pivotal when aiming to obtain a high quality biomass, and the development of an economically sustainable logistics chain is a prerequisite to foster the residue use. However, information on the mechanical pipeline to adopt to efficiently collect plant residues for bioenergetics purposes and maintain their quality are scarce, especially in term of the efficiency of the equipment. To overcome some of the technical barriers related to this process, an innovative prototype (PC50) for pruning collection and chipping was developed [21]. The PC50 chipper was designed by the Italian machine manufacturer ONG S.n.c. with the scientific support of the CREA-IT institute (Italy). Such a prototype is modular and can be used with various types of knives and loading/unloading systems.

In the present study, we reported comprehensive results of the performance and quality of the wood chips produced by the PC50 during the harvesting tests carried out in Germany and Spain on five different fruit species and with variable counter rotating roller speed, loading systems and knives type. The results of different trials on various pruning residuals and under different environmental condition were combined to define the mean effect of two main factors (knife type and loading system) and their interaction on the productivity (especially in term of collection efficiency), and quality of the work.

## 2. Material and methods

### 2.1. Experimental field and pruning biomass characterization

Pruning harvesting trials have been carried out on five orchard species: almond (*Prunus dulcis* (Mill.), peach (*Prunus persica* (L.) Batsch), apple (*Malus domestica* Borkh.), plum (*Prunus domestica* L.) and olive (*Olea europaea* L.) in early spring of 2014 and 2015, in five different sites in Spain and Germany. Before each harvesting test, field shape, field length and field width, and row and plant distances were measured. Prunings were put in swaths ready to be harvested by the chipper model PC50. The characterization of the swaths (in terms of average height and width) and pruning (average basal diameter and length of the branches) were carried out. Table 1 reports the main characteristics of the orchards.

### 2.2. Pruning chipper prototype ONG PC50

The chipper model PC50 was designed by the machine manufacturer ONG s.n.c. of Naldoni Domenico & C., (Castel Bolognese (RA)–Emilia Romagna region–Italy) with the collaboration of the CREA-IT (Council for Agricultural Research and Economics—Research Centre for Engineering and Agro-Food Processing, Monterotondo, Roma, Italy) for windrowing, picking, conveying, chipping, and discharging the pruning biomass directly into a big bag, in a trailer towed by a tractor side-by-side of the machine, or in a tilting box (a container directly installed on the PC50). The installation of the container or big bag system can be interchanged at the farm level according to needs and requires few minutes. Indeed, the peculiarity of the prototype is the versatility and ability to adapt to different farm contexts. This feature of the machine makes it unique and to the best of our knowledge, there are no modular machines in the market with similar characteristics.

The discharge method to employ depends on the collection logistics, and the availability of facilities. Each discharge system has pros and cons as observed by [22]. During the trials two different collection logistic systems were tested (Fig 1):

- Big bag (BB): a breathable plastic bag where to discharge the comminuted biomass;

- Tilting box (TB): a container with a volume of 3 m$^3$.

The chipper PC50 is 1.75 m wide, 3.9 0m long and needs to be towed by a tractor with a minimum power of 45 kW. It represents a novelty because is a modular prototype for pruning collection developed to work to different harvesting logistics and characteristics of prunings, producing also a comminuted product of variable size according to the needs. In fact, PC50 is

**Table 1. Main parameters of orchard species used during the trials and pruning characteristics.**

| Nation | Site | Cropping season | Plant Species | Variety | Age (Years) | Planting pattern (m×m) | Plant density (n ha$^{-1}$) | Pruning branch diameter (mm) | Pruning branch lenght (mm) | Pruning branch moisture content (%) |
|--------|------|-----------------|---------------|---------|-------------|------------------------|-----------------------------|------------------------------|----------------------------|-------------------------------------|
| Spain | Alcaniz | 2014 | Almond | Guara | 8 | 7.0×7.0 | 204 | 30±5** | 1300±430 | 31.6 |
| | Fraga | 2014 | Peach | Paraguayo | 3 | 5.0×3.0 | 666 | 21±8 | 1304±569 | 39,9 |
| | Caspe | 2015 | Olive | Arbechina | 8 | 4.0×1.8 | 1389 | 20±11 | 1592±419 | 33,7 |
| Germany | Postdam | 2015 | Apple | Pinova | 6 | 3.3×1.2 | 2525 | 17±10 | 1103±458 | 43.3 |
| | | | Plum | n.a.* | 10 | 4.0×3.2 | 781 | 33±17 | 2410±723 | 40,8 |

*, n.a., information not available

**, mean ± s.d, n = 50.

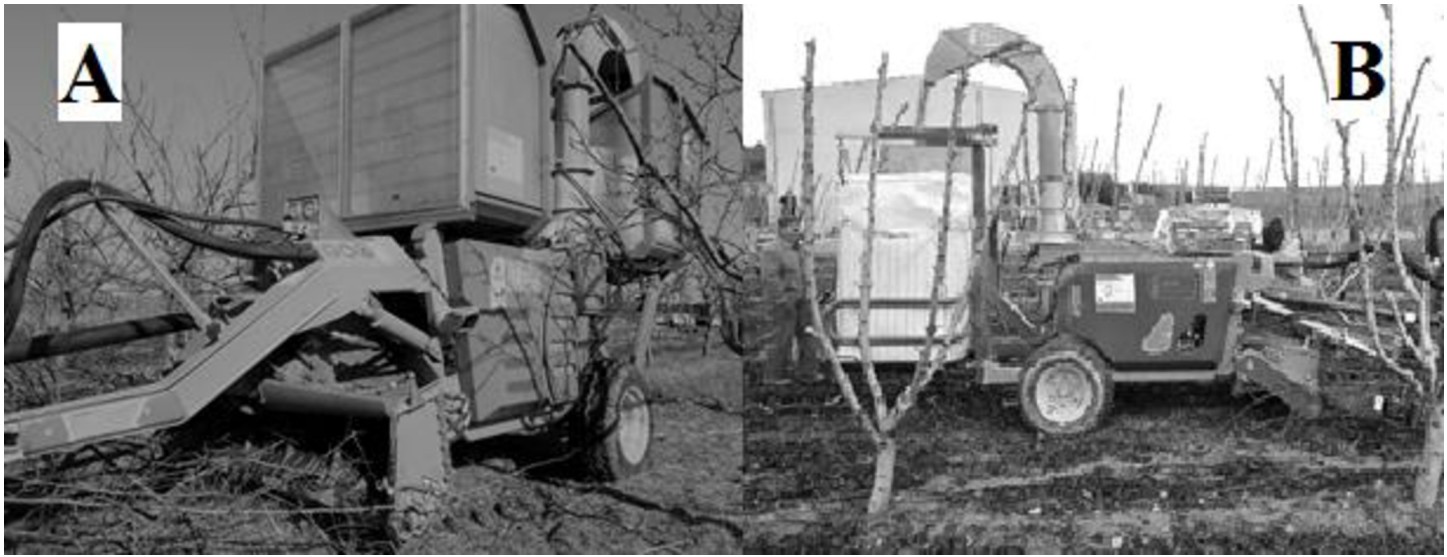

**Fig 1. Naldoni prototype PC50 at different configuration.** A) tilting box and B) big-bag during the harvesting.

equipped with an innovative chipping system, based on an auger that can mount two different type of knives: discontinuous hoe shaped blades and continuous helicoidal knife (Fig 2).

The set of continuous helicoidal knives (c_HEL) consists of two elements positioned consecutively in the direction of rotation of the auger. The two elements work as a single blade that cuts from left to right creating a "scissor effect" by the blade and counter blade. This permits to obtain a clean cut of the wood, reducing the use of the tractor power.

The discontinuous hoe shaped knives set (d_HSK) includes six elements that perform a cut similar to the drum chipper of a forage harvester [23, 24]. The hoe blades of the d_HSK system are mounted offset from each other to cut one blade at a time. Together with the cutting angle

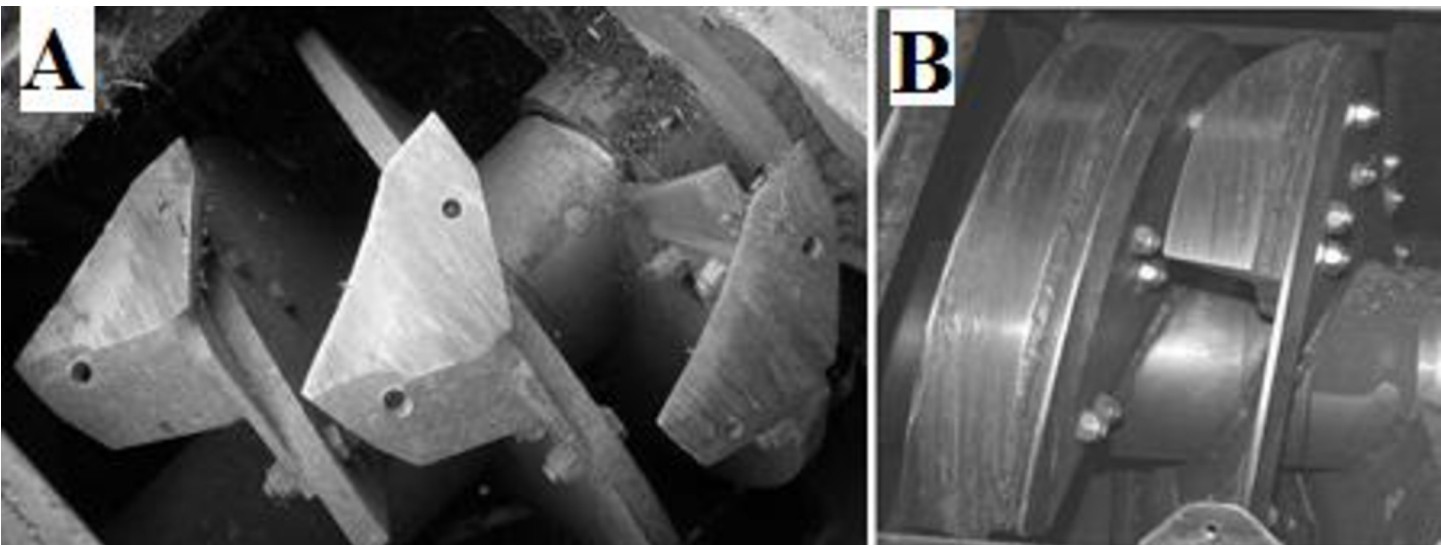

**Fig 2. Types of knife of Naldoni prototype that used during the trial in the field.** A) discontinuous hoe shaped knife (d_HSK); and B) continuous helicoidal knife (c_HEL).

of the blades, this system has been figured out to reduce the power absorption of the tractor to cut the pruning and consequently to reduce fuel consumption.

Once collected from the ground by the pick-up system, pruning is conveyed by the feeding system to the chipping system. A central chain pulls the pruning towards two counter-rotating toothed rollers (CRR) whose rotation speed can be controlled by the tractor operator. The higher the speed of CRR, the greater the length of the woodchips produced. European stakeholders identified the 'P45' woodchip particle size class as the best suited to their needs [25]. Such a distribution identifies a biomass for fuel with 60% of the total mass with particles between 45 and 3.15 mm, a coarse fraction should be less than 10% of the total mass, and the maximum length of the particles less than 350 mm. For this reasons, during the trials different speed of the counter rotating rollers were tested to highlight the relationship between the CRR speed and woodchip characteristics [26].

## 2.3. Evaluation of Naldoni prototype performance

**2.3.1. Working time study and fuel consumption.** PC50 prototype performance was evaluated by measuring the working time according to the methodology reported by [27] in accordance to the Commission Internationale de l'Organisation Scientifique du Travail en Agriculture (C.I.O.S.T.A.) and the Italian Society of Agricultural Engineering (A.I.I.A.) [28]. Fuel consumption was measured for both the productive (chipping phase) and unproductive works (turns, idle travel at field ends, unloading of wood chips, adjusting equipment, and stops with the engine running). Fuel consumption was measured by filling the tank at the end of the test. In the case of consumption of productive work only, chipping tests were done on five rows of varying length (depending on the experimental field). It is important to highlight that the method of evaluating fuel consumption used, although on the one hand very common because easy to apply in the field, on the other hand it has a low accuracy as indicated by other studies, especially when the quantities to be measured are small, and the measurement error is difficult to evaluate. The recorded parameters were used to provide several info, such as harvested yield (actual product harvested) (HY) and total yield (i.e. the pruned material either collected or not) [Mg ha$^{-1}$] (TY), effective and theoretical working capacities both expressed as h ha$^{-1}$, material capacity [Mg h$^{-1}$] (MC), fuel consumption [either as per unit area, i.e. l ha$^{-1}$, and unit biomass collected, i.e. l Mg$^{-1}$]. The harvested yield of each experimental site was obtained measuring all the big bags in the field using a tractor with fork connected to a dynamometer made by PCE Italia Srl (CS 1000N model -range of measurement 1000 kg and sensitivity 0.2 kg). The loose comminuted product discharged in the tilting box was measured using a farm scale. Total yield represents the potential harvestable biomass and it was obtained as a sum of the harvesting yield and the harvesting losses.

**2.3.2. Harvesting losses.** The amount of pruning not collected by the machine during the trials and that remained on the ground, i.e the biomass losses during harvesting, was measured. Harvesting losses have been estimated collecting and weighing the material remained on the ground after the harvesting phase on five random chosen plots. Each plot had an area of 5 m length × inter-row of the orchard (m).

## 2.4. Characterization of pruning biomass chipped

**2.4.1. Apparent bulk density (ABD).** Apparent bulk density [ABD] of wood chips produced by the chipper P50 was measured according to [29] using a normalized cylinder with an internal volume of 26 dm$^3$ and expressed as kg m$^{-3}$. Five samples of wood chips per each blade type were used to fill the cylinder loosely to the top and weighed with a KERN GmbH dynamometer (CH 50K50 model—range of measurements 50 kg and sensitivity 50 g).

**2.4.2. Moisture content.** The moisture content (MC) is the main factor affecting the calorific value of the fuel and then the combustion. Thus, the moisture content of each type of pruning was determined according to [30]. Five samples of wood chips about 500 g each were taken from the pruning residues chipped with two different blades used (hoe shaped blades and helicoidal knife). To prevent the drying, all samples were stored inside the polyethylene bags.

**2.4.3. Particle size distribution (PSD).** The particle size distribution (PSD) is an important parameter and has a crucial role both for the fluidity of the boiler supply and for an efficient combustion. A good quality of the chip is possible achieved when most of the woodchip is within the medium class and the amount of oversized particles and dust is low. The particle size distribution was determined according to [31]. For each trial, the PSD of the comminuted product was carried out to a sample of 20 kg of pruning biomass using a mechanical vibrator sieve Analysette mod. 18 (Fritsch GmbH, Idar-Oberstein, Germany) in accordance with [32]. Through the seven sieves of the vibrating sieving system it was possible to separate the wood-chips in the following particle size classes: <3.15 mm; 3.15–8 mm; 8–16 mm; 16–45 mm; 45–63 mm; 63–100 mm, 100–120 mm; 120–350 mm.

## 2.5. Statistical analysis

The analysis of variance was performed according to an unbalanced randomized block design model. Such a model was applied by means of a general linear mixed model (Glimmix procedure in SAS/STAT 9.2 statistical package; SAS Institute Inc., Cary, NC, USA). The model used was similar to that shown in the supplementary material in [33] with both a description of the procedure and the SAS procedure applied. In the analyses of the machine performances, fixed factors studied were the loading system of the chipped residues (either in a big bag or a tilting box) and the knives type used for the chipping (either a discontinuous hoe shape knives (d_HSK) or a continuous helicoidal shaped (c_HEL) (Fig 2A and 2B). The loading system × knife type interaction was included among the fixed factors. Random variables included the plant species, the number of replicate (corresponding to a row of variable length) nested into the plant species, and if such a row was the first performed in each test to take into account that the tractor may have had a cold engine (Table 2).

Quality of the chipped material in term of size distribution, moisture and apparent bulk density were analyzed separately by the speed of the counter rotating rollers and the type of the knives tested. In particular, a direct comparison between knives types within the same species and at a given speed of the counter rotating rollers was available only for the peach, thus the

**Table 2. Variables related to the machine performances, number of replicate per treatment: Loading system [LS] and Knife type [KT].**

| Country | Experimental Site | Cropping season | Plant Species | LS* | KT* | Theoretical work capacity (h ha-1) | Actual work capacity (h ha-1) | Field efficiency (%) | Material Capacity (t h⁻¹) | Losses (Mg fw ha⁻¹) | Havested yield (HY) (Mg ha⁻¹) | Total yield (TY) (Mg ha⁻¹) | Collection efficiency (%) | Fuel consumption (l ha⁻¹) | Fuel consumption (l HY Mg⁻¹) |
|---|---|---|---|---|---|---|---|---|---|---|---|---|---|---|---|
| Spain | Alcaniz | 2014 | Almond | BB | d_HSK | 11 | 11 | 11 | 5 | 4 | 5 | 5 | 5 | 5 | 5 |
| | | | | TB | d_HSK | 8 | 8 | 8 | 5 | 5 | 5 | 5 | 5 | 5 | 4 |
| | Fraga | 2014 | Peach | BB | d_HSK | 10 | 10 | 10 | 3 | 5 | 3 | 3 | 3 | 3 | 3 |
| | | | | | c_HEL | 4 | 4 | 4 | 2 | 5 | 2 | 2 | 2 | 2 | 2 |
| | Caspe | 2015 | Olive | BB | c_HEL | 14 | 14 | 14 | 8 | 5 | 8 | 8 | 8 | 8 | 8 |
| Germany | Postdam | 2015 | Apple | TB | c_HEL | 5 | 5 | 5 | 3 | 5 | 3 | 3 | 3 | 3 | 3 |
| | | | Plum | BB | c_HEL | 3 | 3 | 3 | 3 | 5 | 3 | 3 | 3 | 3 | 3 |
| | | | | TB | c_HEL | 8 | 8 | 8 | 2 | 9 | 2 | 2 | 2 | 2 | 2 |

* BB, Big Bag; TB, Tilting Box; d_HSK, hoe shaped knives, c_HEL, Helicoidal knives.

Table 3. Variables related to the product quality, number of replicate per treatment: Knife type [KT], Rotation speed [RS], and Loading system [LS].

| Plant Species | Cropping Season | KT** | RS** | LS** | Replicate per each size fraction | Moisture [%] | ABD* [kg m⁻³] |
|---|---|---|---|---|---|---|---|
| Almond§ | 2014 | d_HSK | V4 | TB | 4 | 5 | n.m§§ |
| Peach | 2014 | c_HEL | V3 | BB | 4 | 5 | 5 |
| | | d_HSK | V3 | BB | 4 | 7 | 5 |
| Olive | 2015 | c_HEL | V4 | BB | 4 | n.m | n.m |
| | | | V5 | BB | 4 | n.m | n.m |
| Apple or Plum§ | 2015 | c_HEL | V2 | TB | 4 | n.m | 5 |
| | | | V3 | TB | 4 | n.m | 5 |
| | | | V4 | TB | 4 | n.m | 5 |
| | | | V5 | TB | 4 | n.m | 5 |
| | | | Vmax | TB | 4 | n.m | 5 |

\* apparent bulk density.

\*\* BB, Big Bag; TB, Tilting Box; d_HSK, hoe shaped knives, c_HEL, Helicoidal knives; V2, 0.85 Hz (51 rpm); V3, 1.23 Hz (74 rpm); V4, 1.65 Hz (99 rpm); V5, 2 Hz (120 rpm); Vmax, 2.4 Hz (144 rpm).

§ in almond, 4 replicates with 5 subreplicates each were available. For Apple or Plum, replicates are indicated per species, i.e.4 = 4 replicates in apple + 4 in plum.

§§ n.m, not measured.

role of type of knife on the chipped material quality was analyzed only for this species, and replicate was the random factor (Table 3).

Various speed of CRR were tested in many species only when using the continuous helicoidal shaped knives, so that only such data were used for this comparison (Table 3). In such an analysis, random factors were the loading system, the plant species, the replicate and the subreplicate within a replicate, when applicable.

In both analyses, unbiased estimates of variance and covariance parameters were assessed by restricted maximum likelihood (REML). Denominator degrees of freedom of each error were estimated by Kenward–Roger approximation (according to which null covariance parameter do not contribute to degrees of freedom of the model) and least square means of the treatment distributions were computed. LSmeans were used instead of the means to compare for the treatment effects. Briefly, LSmeans are estimation of an estimated population mean from an unbalanced data population. The LSMEANS estimates the corresponding means in a factor level after taking into account both the distribution of the values of the variable within the level (fixed and random factors), and the distribution of the values within the population. In the results sections, data were provided both as LSmeans in figures and arithmetic means in supplemental materials, along with each standard error estimation or computation, respectively. Differences among means were compared by applying tukey-kramer grouping at the 5% probability level to the LSMEANS p-differences. When denominator degrees of freedom were not constant and in the presence of heteroscedasticity, "ADJDFE = ROW" statement was used to adjust for multiple comparisons. The complete raw datasets were provided as a S1 and S2 Tables for performance and quality data, respectively.

## 3. Results and discussion

Pruning harvesting supply chain includes a first step of swath formation that can be carried out using rakes separated or integrated to the harvester machine, and a second phase of biomass collection. In this second step the pruning can be processed by shredders, chippers or

balers. The machines for pruning harvest available on the market that are currently the most diffused use the shredding technology, although, technology involving chipping results in a better quality comminuted product [18].

Vineyard and olive groves represent highly widespread perennial crops in Europe and for this reason received great attention in testing pruning shredders [22, 34–39] and experimental trials are also beeing conducted for other species [22, 40].

As observed by [18], chippers, in contrast to shredders, are newly built solutions for pruning and different number and shape of knives are used by chippers to comminute the pruning. According to Spinelli et al. [41], comminution with knives can result in 30–80% (average 50%) more productivity than using the hammers, while requiring 15–30% (average 24%) less energy per unit product depending on feedstock type. Also the fuel consumption is affected by the comminution devices. Spinelli et al. [41] reported 30–75% (average 52%) higher fuel consumption of hammer compared to knives configuration. On the one hand, chippers process the wood with sharp blades causing the biomass to be cut in more homogeneous pieces [18, 42]; on the other hand, knives are more prone to be damaged in the presence of dirt, stones or metals that may seriously compromise the cutting system if collected during the harvesting, affecting productivity, performance and machine costs. Unfortnatly, presence of stones and soil coarse fraction is frequent in orchards. These aspects have been well analyzed [43] observing a reduction of productivity of 20% in the presence of knife wear.

Facello et al. (2013) reported even greater chipper productivity loss (50%) with worn knives [44]. Knives wear also affected both the fuel consumption and woodchip quality; fuel consumption increased 1.4–2.8 times, whereas the reduction of woodchip quality was due to a marked increase in small chips (8–3 mm) and fine particles (<3 mm), and a reduction in large chips (45–9 mm) [44]. Indeed, particle shape is strongly related to knife sharpness [45]. Among the various tests performed with the PC50, two types of knives (d_HSK, hoe shaped knives, c_HEL, Helicoidal knives) were compared and the influence of knife wear was not verified. It should also be pointed out that, to the best of our knowledge, there are no studies in the literature on the evaluation of productivity of pruning harvesting machines with helicoidal cutting system, and the analysis of the various interaction among and this research represent a novelty of this specific field. Only Wegener and Wegener [46] reported the basic principle of helical chippers, and Wegener and Wegener [47] described three prototypes used for SRF harvesting or shrubland clearing equipped with the helical chipper system. However, these studies did not report harvesting performance data nor the quality of the wood chips produced.

## 3.1. Machine performance

The variability of technical parameters such as working width, harvesting speed, container capacity, and field conditions (species, planting pattern, amount of biomass available) determines a strong variability of the performance and cost. As already mentioned, although the literature reports several experiences of pruning harvesting by shredders and their performance, there are few documented experiences of pruning harvesting carried out by chippers. In general, chippers available in the market showed a productivity of 1.99 t h$^{-1}$ (0.86 ha h$^{-1}$) to 2.8 t h$^{-1}$ in vineyard [18] and between 1.07 to 1.26 t h$^{-1}$ in olive groves [48]. Spinelli and Picchi [36] reported the productivity obtained on olive groves by two different machines, both designed for industrial pruning collection, one combined with a powerful agricultural tractor, the other with a dedicated self-propelled machine: the productivity found here was significantly higher compared the other studies already cited (3.0–9. 4 t h$^{-1}$).

Table 4 reports the results obtained during the performance harvesting tests carried out with the PC50 in Spain (almond, peach, olive) and Germany (apple and plum).

**Table 4. Main results obtained during the performance harvesting tests carried out with the PC50 in Spain and Germany.** See Table 5 for the analysis of the main factor loading system (LS, either BB, Big Bag; or TB, Tilting Box) and knives type (KT, either d_HSK, discontinous hoe shaped knives, c_HEL, continous helicoidal knives) and their interaction by means of a general linear mixed model adapted to unbalanced datasets where plant species and Counter Rotating Rollers (CRR: V4 (1.65 Hz) for almond and olive and V3 (1.23 Hz) for Peach, Apple and Plum) were considered random factor. Data are means±standad error.

| Species | LS* | KT* | Theoretical working capacity (h ha$^{-1}$) | Actual working capacity (h ha$^{-1}$) | Field efficiency (%) | Material Capacity (t$_{fm}$* h$^{-1}$) | Losses (t$_{fm}$ ha$^{-1}$) | Havested yield (HY) (t$_{fm}$ ha$^{-1}$) | Total yield (t$_{fm}$ ha$^{-1}$) | Collection efficiency (%) | Fuel consumption | |
|---|---|---|---|---|---|---|---|---|---|---|---|---|
| | | | | | | | | | | | (l ha$^{-1}$) | (l t$_{fm}$$^{-1}$) |
| Almond | BB | d_HSK | 0.41±0.02 | 0.64±0.04 | 64.5±5.5 | 0.61±0.04 | 0.06±0.01 | 0.38±0.01 | 0.43±0.01 | 87.2±0.4 | 4.86 ±0.83 | 12.86 ±2.17 |
| | TB | d_HSK | 0.48±0.05 | 0.83±0.06 | 57.4±6.3 | 0.38±0.04 | 0.05±0.03 | 0.37±0.04 | 0.39±0.04 | 94.8±2.2 | 5.84 ±1.69 | 11.18 ±3.05 |
| Peach | BB | d_HSK | 2.45±0.14 | 3.30±0.16 | 74.2±4.1 | 0.76±0.04 | 0.51±0.11 | 2.48±0.05 | 2.89±0.04 | 86.0±1.4 | 18.51 ±1.41 | 7.47 ±0.65 |
| | BB | c_HEL | 1.38±0.06 | 1.82±0.19 | 75.8±7.7 | 1.13±0.19 | 0.31±0.07 | 2.05±0.01 | 2.26±0.07 | 90.6±2.3 | 23.26 ±0.16 | 11.36 ±0.16 |
| Olive | BB | c_HEL | 2.92±0.14 | 3.64±0.19 | 80.0±2.8 | 1.13±0.05 | 1.04±0.26 | 3.22±0.23 | 4.11±0.24 | 77.9±1.6 | 27.41 ±2.05 | 8.74 ±0.71 |
| Apple | TB | c_HEL | 1.54±0.06 | 1.82±0.15 | 84.9±4.2 | 1.55±0.13 | 0.69±0.06 | 2.18±0.02 | 2.78±0.05 | 78.4±0.8 | 17.67 ±2.11 | 8.13 ±1.01 |
| Plum | BB | c_HEL | 4.22±0.69 | 4.86±0.43 | 86.9±6.2 | 1.11±0.33 | 0.87±0.13 | 5.11±1.52 | 5.88±1.53 | 84.6±4.5 | 16.40 ±0.26 | 4.09 ±1.53 |
| | TB | c_HEL | 4.48±0.75 | 5.42±1.00 | 82.7±5.2 | 1.05±0.81 | 0.88±0.09 | 9.55±0.01 | 10.16 ±0.01 | 94.0±0.1 | 13.51 ±0.68 | 1.42 ±0.07 |

* fm indicates fresh matter, i.e. non dried wood material.

The largest amount of available biomass was found in the plum orchard where the harvesting losses resulted lower than the other tests. This result likely affected the field capacity (ha h$^{-1}$) of the harvesting system: the more biomass available, the lower field capacity. This likely depended on a slower working speed of the machine compared to the other species. Although, regarding the other parameters related to harvesting performance, such as MC and fuel consumption, it is rather complicated to understand how they varied with respect to the loading system of the product used, the type of knives mounted on the chipper, the characteristics of the pruning (diameter, wood hardness, humidity) and the biomass available (Table 4). For this reason, the statistical analysis adopted carefully took into account the influence that these parameters have on the outcome. Same issues regards the quality of the wood chips produced in terms of particle size distribution and apparent bulk density.

In Table 5 results of the general linear mixed analysis of the machine performance in term of loading system, knife type and their interaction are reported. The role of the treatments (i.e. the machine management option) scarcely affected the variables measured (Table 5).

Such a paucity of effects may be due to both a scarce effect of the treatments and high covariate effects (S3 Table) and especially that of the plant species, which covariance parameter estimate was frequently high (although with a high standard error estimate).

In particular, only the Field efficiency and Material Capacity varied by the treatments. Material capacity of the chipper varied by the knife type: the test carried out with the machine equipped with continuous knife (1.26±0.18 t h$^{-1}$) resulted 97% higher than with discontinuous hoe shape knives (0.64±0.20 t h$^{-1}$).

Similarly, data analysis showed that the interaction between KT and LS had significant statistical effect on the field efficiency parameter. In particular, c_HEL allowed a higher field

**Table 5. Results of the general linear mixed analysis of the machine performance in term of Loading system (LS), knife type (KT) and their interaction.** Factors and interaction at p<0.05 are shown in bold. Replicates are shown in Table 2, the covariates effect is shown in the S3 Table.

| | | LS | | | KT | | | LS×KT | | |
|---|---|---|---|---|---|---|---|---|---|---|
| | | Den DF | F | p | Den DF | F | p | Den DF | F | p |
| Effective time | [h ha-1] | 3.11 | 1.07 | 0.37 | 3.17 | 0.04 | 0.85 | 3.10 | 0.94 | 0.40 |
| Operation time | [h ha-1] | 3.12 | 0.90 | 0.41 | 3.19 | 0.12 | 0.75 | 3.11 | 0.53 | 0.52 |
| Field efficiency | [%] | 45.60 | 1.10 | 0.30 | **47.89** | **18.50** | **< .001** | **43.40** | **7.30** | **0.01** |
| Material Capacity | [t h-1] | 3.28 | 0.04 | 0.85 | **3.71** | **11.62** | **0.03** | 3.26 | 3.43 | 0.15 |
| Losses | [t FW ha-1] | 3.30 | 0.96 | 0.39 | 3.41 | 0.04 | 0.85 | 3.19 | 1.07 | 0.37 |
| Havested yield (HY) | [t ha-1] | 3.01 | 1.88 | 0.26 | 3.01 | 1.12 | 0.37 | 3.01 | 1.90 | 0.26 |
| Total yield | [t ha-1] | 3.00 | 1.87 | 0.26 | 3.00 | 1.00 | 0.39 | 3.00 | 1.96 | 0.26 |
| Collection efficiency | [%] | 3.56 | 0.80 | 0.43 | 4.63 | 0.00 | 0.95 | 3.35 | 0.87 | 0.41 |
| Fuel consumption | [l ha-1] | 3.42 | 0.42 | 0.56 | 3.92 | 0.52 | 0.51 | 3.33 | 0.81 | 0.43 |
| | [l (t HY)-1] | 4.68 | 2.31 | 0.19 | 6.31 | 0.02 | 0.88 | 4.34 | 1.75 | 0.25 |

efficiency than d_HSK (on average 88.6±9.1% vs. 71.1±9.3%), but such efficiency also varied by the loading system (Fig 3).

The other variables measured did not vary according to the treatments. Total yield (i.e. the pruned material either collected or not) was 0.37 t ha$^{-1}$ in almond, 2.18 t ha$^{-1}$ in apple, 3.22 t ha$^{-1}$ in olive, 2.31 t ha$^{-1}$ in peach and 6.88 t ha$^{-1}$ in plum. Out of these yields, losses (that is the pruned material not harvested by the machine) was 13.1%, 24.8%, 25.2%, 15.5%, and 11.5%, respectively (computed by considering the mean harvested yields and not the relative LSMEANS). In turn, this consisted in a collection efficiency (i.e. the ratio between the collected and total yields) of, respectively, 91.0%, 78.4%, 77.9%, 87.8%, and 88.4%. Please note that the sum of losses and collection efficiency is higher than 100% due to unbalancing of the

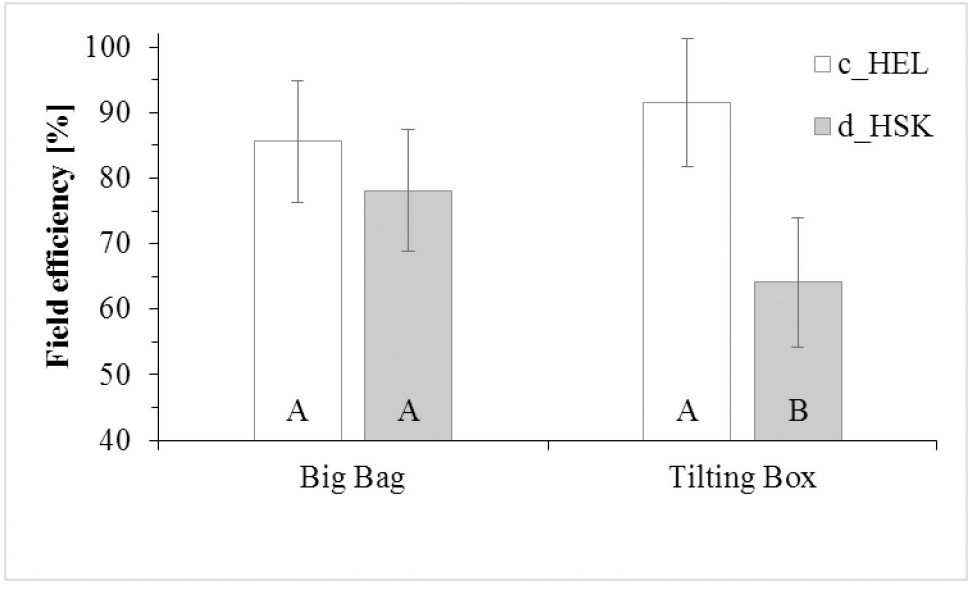

**Fig 3. Field efficiency (%) of the wood chipper prototype equipped with a big bag or tilting box and with either continuous helicoidal knifes (c_HEL) or discontinuous hoe shaped knife (d_HSK).** Bars are the LSmeans ±standard error estimate. Bars with a letter in common cannot be considered different according to the tukey-kramer grouping applied at the 0.05 of the LSmeans p differences.

measurements. Collection efficiency was measured on the row harvesting procedure, and thus is more representative than losses, measured in subsamples.

Fuel consumption expressed either per unit of area (l ha$^{-1}$) dramatically varied among plant species and this was likely due to differences in the tractor used and yields per unit of area. In particular, fuel consumptions were 5.35 l ha$^{-1}$, 17.67 l ha$^{-1}$, 27.41 l ha$^{-1}$, 20.41 l ha$^{-1}$, and 15.25, l ha$^{-1}$ in almond, apple, olive, peach and plum, respectively. When measured per ton of harvested yield, such data were 12.11 l t$^{-1}$, 8.13 l t$^{-1}$, 8.74 l t$^{-1}$, 9.02 l t$^{-1}$, 3.02 l t$^{-1}$.

Despite the lack of a general effect in most variables, some differences detected by the exact LSMEANS p-differences were lost when a conservative Tukey-Kramer grouping (i.e. the "lines" display by the SAS procedure) was used (S4 Table). These differences occurred in the theoretical working capacity of BB d_HSK vs. BB c_HEL (1.99±1.12 h ha$^{-1}$ vs. 1.00±1.12 h ha$^{-1}$, respectively) and actual working capacity of the same pair-wise comparison (2.45±1.35 h ha$^{-1}$ vs. 1.40±1.12 h ha$^{-1}$, respectively). This difference corroborates the hypothesis that variability among trials can have affected the general outcome. And indeed, BB d_HSK showed 28% harvested and 31% total yield more than BB c_HEL when analyzing the pairwise comparison (S4 Table).

## 3.2. Role of the type of knives on the chipped material quality

The role of the type of knives on the quality of the chipped material was studied exclusively in peach where a comparative trial was carried out in similar conditions (tractor type, field and pruning characteristics). No differences by the knives used were seen for the size fraction higher than 45 mm (Table 6). On the other hand, the use of d_HSK resulted in a 50.7% increase in wood chips falling into the 16 to 46 mm particle size class compared to c_HEL. In general, 77.9% in d_HSK and 92.7% in c_HEL of the wood particles were < 45 mm.

In contrast, the use of d_HSK resulted in the production of 33% less wood chips smaller than 16 mm in size. In particular, when compared with c_HSD, the use of d_HSK resulted in a reduction of wood chips belonging to particle size classes < 3.15 mm, 3.15 to 8 mm, and 8 to 16 mm, by 26.3%, 41.3% and 28.4%, respectively. However, these differences in particle size of the wood chips did not result in variation of the ABD between the two KT. At the same time, d_HSK consisted in drier samples than c_HEL, despite differences were negligible (1.27% of absolute moisture difference). However, as reported by Wegener and Wegener [47] "moisture content is important in the chipping process because it acts as a sort of lubricant between the

**Table 6. Quality of the chipped material in peach at varying the knife type (d_HSK, hoe shaped knives, c_HEL, Helicoidal knives).** Quality traits were the size distribution of the material, the mean moisture and the apparent bulk density (ABD). Fractions varying at p>0.05 are shown in bold.

|  | F | P | c_HEL Lsmeans±s.e.e. | d_HSK LSmeans±s.e.e. |
|---|---|---|---|---|
| Size (mm): 120 to 350 | 1.45 | 0.3154 | 5.01±1.89 | 8.22±1.89 |
| Size (mm): 100 to 120 | no data of this fraction | | | |
| Size (mm): 63 to 100 | 4.55 | 0.1228 | 1.03±3.43 | 9.89±3.43 |
| Size (mm): 45 to 63 | 2.09 | 0.2443 | 1.30±1.34 | 4.04±1.34 |
| Size (mm): 16 to 45 | **48.95** | **0.006** | **18.87±2.34** | **28.44±2.34** |
| Size (mm): 8 to 16 | **14.82** | **0.031** | **31.71±1.98** | **22.71±1.98** |
| Size (mm): 3.15 to 8 | **185.66** | **0.0009** | **28.59±2.24** | **16.78±2.24** |
| Size (mm) < 3.15 | **24.00** | **0.0163** | **13.48±2.37** | **9.93±2.37** |
| Moisture (%) | **8.59** | **0.0428** | **60.09±0.33** | **58.82±0.28** |
| ABD (kg m-3) | 0.81 | 0.4197 | 222.41±10.75 | 209.36±10.75 |

work piece and the blade, which effectively reduces the friction". Also, it is important to highlight that the current trend is to carry out the drying of pruning directly in the field before they are harvested in order to reduce transport costs, and allow the leaves to fall in the field (in the case of the olive tree) due to the high ash content, and low energy content.

## 3.3. Role of the counter rotating roller speed on the chipped material quality

CRR speed scarcely affected the quality of the material (Table 7), however, according to our expectation, such effect mostly consisted in a reduction of the shorter sized fractions (<8 mm). IA higher speed of the feeding rollers corresponded to a faster passage of the pruning branches beyond the cutting point of the chipper system. Consequently, the pruning was cut in longer length pieces. Increasing the speed from 0.85 Hz (V2) to 2.4 Hz (Vmax) consisted on average on a reduction of 38.7% (relative variation) of the fractions sized <8 mm, which passed from 39.6% of the samples to 24.3%. Such reduction also consisted in mild increases (frequently not significant) of the fractions > 16 mm.

It is important to highlight that the characteristics of wood chips are influenced by several factors and the use of helical knives for pruning harvesting will require more studies especially regarding the feeding system and its effect on the chipping phase with helical knives. In fact, Wegener and Wegener [46] in their study about the process of wood chip formation by helical chippers stated that that chip length is predominantly influenced by the geometric characteristic of the chipper (cone angle of the rotor and the wedge angle of the blade), but also that the infeed angle can have a significant influence on chip quality. For this reason, Wegener and Wegener [49] emphasized the need for further studies to understand the extent to which the infeed angle of the material enters the chipping chamber can influence the quality of the wood chips produced, as well as the role that the direction of the wood fibers play with respect to the cutting point. Wegener and Wegener [49] also observed that when an infeed parallel to the rotor's axis was used, this resulted in an exceptionally high degree of fraying and a lack of clean

**Table 7. Quality of the chipped material at varying the counter rotating roller speed in the Helicoidal knives (c_HEL) treatments only.** Treatments with Hoe shaped knives (d_HSK) were not included in the analysis because not direct comparison (i.e. at least 2 different speed in the same species) were available. Quality traits were the size distribution of the material and the apparent bulk density (ABD). The mean moisture was not available. Fractions varying at p>0.05 are shown in bold. Within each fractions, LSmeans with a letter in common can't be considered different according to a lines-display of the ranked p-differences adjusted by Tukey Kramer for multiple comparisons. The estimated denominator degrees of freedom (DEN DF) after the Kenward Roger correction is shown, along with F and p values.

| Wood chip particles class | | | | V2* | V3 | V4 | V5 | Vmax |
|---|---|---|---|---|---|---|---|---|
| | **Den DF** | **F** | **p** | LSmeans±s.e.e. | LSmeans±s.e.e. | LSmeans±s.e.e. | LSmeans±s.e.e. | LSmeans±s.e.e. |
| Size (mm): 120 to 350 | 42.80 | 0.38 | 0.821 | 4.08±2.91 | 5.69±2.42 | 4.75±2.44 | 5.42±2.44 | 7.78±2.91 |
| Size (mm): 100 to 120 | **46.05** | **5.98** | **0.001** | **0.00±0.19B** | **0.01±0.16B** | **0.00±0.16B** | **0.00±0.16B** | **0.90±0.19A** |
| Size (mm): 63 to 100 | 45.40 | 0.35 | 0.840 | 2.51±2.71 | 3.45±2.47 | 3.69±2.51 | 4.50±2.51 | 4.74±2.71 |
| Size (mm): 45 to 63 | 46.04 | 1.00 | 0.417 | 2.42±1.32 | 1.48±1.08 | 1.77±1.09 | 3.42±1.09 | 3.69±1.32 |
| Size (mm): 16 to 45 | 46.06 | 1.65 | 0.178 | 28.67±6.97 | 24.88±6.84 | 27.19±6.84 | 31.77±6.84 | 29.45±7.06 |
| Size (mm): 8 to 16 | 32.79 | 0.51 | 0.727 | 28.65±5.08 | 29.21±4.21 | 30.95±4.24 | 31.45±4.24 | 28.69±4.38 |
| Size (mm): 3.15 to 8** | **13.85** | **7.28** | **0.002** | **29.14±5.56A** | **25.14±4.18AB** | **23.48±4.21AB** | **16.25±4.21C** | **17.78±4.33D** |
| Size (mm) < 3.15 | **46.40** | **3.83** | **0.009** | **10.46±1.51AB** | **10.66±1.36A** | **7.33±1.36AB** | **6.35±1.36B** | **6.49±1.67AB** |
| Moisture (%) | no data of this variable | | | | | | | |
| ABD (kg m-3) | **44.20** | **16.93** | **< .0001** | **296.19±18.49A** | **258.70±18.01B** | **279.04±18.49A** | **252.19±18.49B** | **246.23±18.49B** |

* V2, 0.85 Hz (51 rpm); V3, 1.23 Hz (74 rpm); V4, 1.65 Hz (99 rpm); V5, 2 Hz (120 rpm); Vmax, 2.4 Hz (144 rpm).

** The LINES display does not reflect all significant comparisons. The following additional pairs are significantly different: (V3,Vmax), (V3,V5), (V4,V5). Letters were thus rebuilt according to the direct p-difference pairwise test after a Tukey-Kramer adjustment for Multiple Comparisons.

division between individual wood chips. In the case of the prototype PC50 the pruning collected from the ground by the pickup system are transported to the feeding rollers with variable arrangements depending on the length of the branches and their position on the windrow. For this reason, some branches may arrive at the CRR parallel to the axis of the chipper and other branches perpendicular to it. Thus, according to the results presented by [49], the production of a wood chip from pruning with uniform characteristics using a helical chipper is extremely difficult unless a system is devised that positions the branches orthogonally to the axis of the helical chipper.

Furthermore, Wegener and Wegener [46] reported that the characteristics of the material influence the chip length and especially its thickness that appears to be negatively correlated with the material's shear strength. Wood moisture is another parameter that influence the characteristics of wood chips. Wood with higher moisture content leads to larger chip length in average [49]. In general, increasing the counter rotating roller speed brought to reductions in the apparent bulk density, which reduced by 50 kg m$^{-3}$ (corresponding to a 16.9% reduction) from 0.85 Hz (V2) to 2.4 Hz (Vmax). This result agrees with Rackl and Günthner [50] in that a larger woodchip size due to a higher feed roller speed also results in a reduction of ABD.

## 4. Conclusions

In this study, we carried out a comprehensive analysis of the performance and quality of the wood chips produced by an innovative prototype for pruning collection and comminution from tests in five different fruit species and with different machine settings: loading systems and knives type. Using continuous helicoidal knives consisted in a higher productivity (i.e. more product in the unit time) compared to discontinuous hoe shape knives. The minimum field efficiency of the machine was observed when d_HSK knife was used and the product was unloaded on the tilting box. In all other combinations (LS×KT), no significant differences in performance were observed. Different results were obtained when comparing the particle size distribution of the products obtained with the two different cutting systems, with discontinuous hoe shape knivesdelivering bigger chips and reducing of the 26.3% the production of <3.15mm compared to continuous helicoidal knives. Lastly, increasing CRR speed consisted on average on a reduction of 38.7% (relative variation) of the fractions sized <8 mm.

These results have implication in the design of new tools for the pruning collection. In particular, the fuel consumption in this study resulted from 1.42 to 12.86 l fuel (t harvested product)$^{-1}$, depending on the plant species and machine setting. These results imply a high room for the machine improvement when aiming to reduce the energetic cost of the pruning collection, especially if considering their potential energy content.

Some aspects of the collection efficiency remain to be investigated: the effect of the wear of the knives and its variation in the two types of knives was not evaluated here, nor the influence of the hardness of the different pruning processed, which may play an important role in the quality of the work and performance of the chipper and could be used to better set the machine operational conditions.

## Supporting information

**S1 Graphical abstract.**
(TIF)

**S1 Table. Complete row dataset of the "performance" variables used to build the LSmeans.**
(XLSX)

**S2 Table. Complete row dataset of the "quality" variables used to build the LSmeans.** (XLSX)

**S3 Table. Covariance parameter estimates (est.) and relative standard error estimate (s.e. e.) for the analysis of the traits related to the machine performances.** The replicate as a first row or not was nested with each plant species. Plant species treatment englobes the variability of the sites, cropping season and crop management, when the covariance parameter was null (i.e. zero), its standard error was indicated as not estimable (n.e.). Variables with null covariance parameter were retained in the analysis because this allow the Kenward Roger procedure to better estimate the denominator degrees of freedom of the fixed factors. (DOCX)

**S4 Table. Least square means (LSmeans) estimates (est.) and relative standard error estimate (s.e.e.) for the analysis of the machine performances variables and indication of the direct LSmeans p-difference (i.e. Lines display note) when not reflected in the conservative Tukey-Kramer grouping comparison.** (DOCX)

## Acknowledgments

We wish to thank Mr Domenico Naldoni and the employees of ONG snc for their professionalism and cooperation.

## Author Contributions

**Conceptualization:** Alessandro Suardi, Sergio Saia, Vincenzo Alfano.

**Data curation:** Alessandro Suardi, Sergio Saia, Vincenzo Alfano.

**Formal analysis:** Sergio Saia.

**Funding acquisition:** Luigi Pari.

**Investigation:** Alessandro Suardi, Vincenzo Alfano.

**Methodology:** Alessandro Suardi.

**Software:** Sergio Saia.

**Supervision:** Luigi Pari.

**Writing – original draft:** Alessandro Suardi, Sergio Saia, Vincenzo Alfano, Negar Rezaei, Paola Cetera.

**Writing – review & editing:** Alessandro Suardi, Sergio Saia, Vincenzo Alfano, Simone Bergonzoli.

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
