## [Decision Letter · Decision Letter 0]

20 Apr 2021

PONE-D-21-04650

Pruning harvesting with modular towed chipper: scarce role of the machine setting and configuration on performance despite strong impact on wood chip quality

PLOS ONE

Dear Dr. Saia,

Thank you for submitting your manuscript to PLOS ONE. After careful consideration, we feel that it has merit but does not fully meet PLOS ONE’s publication criteria as it currently stands. Therefore, we invite you to submit a revised version of the manuscript that addresses the points raised during the review process.

We look forward to receiving your revised manuscript.

Kind regards,

Yongbo Li

Academic Editor

PLOS ONE

Journal Requirements:

3. During your revisions, please note that a simple title correction is required: to follow correct English language usage, the title should read "Pruning harvesting with modular towed chipper: little effect of the machine setting and configuration on performance despite strong impact on wood chip quality".

Please ensure this is updated in the manuscript file and the online submission information.

4. We note that Figure 1 includes an image of individuals.

As per the PLOS ONE policy (http://journals.plos.org/plosone/s/submission-guidelines#loc-human-subjects-research) on papers that include identifying, or potentially identifying, information, the individual(s) or parent(s)/guardian(s) must be informed of the terms of the PLOS open-access (CC-BY) license and provide specific permission for publication of these details under the terms of this license.

Please download the Consent Form for Publication in a PLOS Journal (http://journals.plos.org/plosone/s/file?id=8ce6/plos-consent-form-english.pdf).

The signed consent form should not be submitted with the manuscript, but should be securely filed in the individual's case notes. Please amend the methods section and ethics statement of the manuscript to explicitly state that the patient/participant has provided consent for publication: “The individual in this manuscript has given written informed consent (as outlined in PLOS consent form) to publish these case details”.

If you are unable to obtain consent from the subject of the photograph, you will need to remove the figure and any other textual identifying information or case descriptions for these individuals.

Reviewers' comments:

Reviewer's Responses to Questions

**Comments to the Author**

1. Is the manuscript technically sound, and do the data support the conclusions?

Reviewer #1: Yes

Reviewer #2: Yes

2. Has the statistical analysis been performed appropriately and rigorously? 

Reviewer #1: Yes

Reviewer #2: Yes

3. Have the authors made all data underlying the findings in their manuscript fully available?

Reviewer #1: Yes

Reviewer #2: Yes

4. Is the manuscript presented in an intelligible fashion and written in standard English?

Reviewer #1: Yes

Reviewer #2: No

5. Review Comments to the Author

Reviewer #1: This paper presents a research on pruning harvesting based on the modular chipper prototype PC50. A statistical analysis was conducted to find out the best combination of knife type, unloading system and speed of the feeding system of the mentioned prototype. This paper is good. However, in my opinion following changes are necessary before publication:

1. Please clarify more why author specifically choose this particular prototype. If possible, present a comparison with existing other modular chipper.

2. Please try to improve the English language.

3. Correct the headings number ?

4. Reference citation should be consistent throughout the paper.

5. Conclusions are needed to be more specific and concise.

Reviewer #2: Some comments are given as follows:

(1) Lack of significant contributions: The contribution of this paper is not significant. The proposed methodology has limited novelty, and the overall impact is not significant enough;

(2) Clarity and Presentation: the paper is not well presented, and there are quite some English writing errors/typos. The authors are suggested to check the whole paper carefully to avoid such errors.

6. PLOS authors have the option to publish the peer review history of their article (what does this mean?). If published, this will include your full peer review and any attached files.

Reviewer #1: No

Reviewer #2: No

---

## [Author Response · Author response to Decision Letter 0]

11 Jun 2021

Response

Reviewer #1: This paper presents a research on pruning harvesting based on the modular chipper prototype PC50. A statistical analysis was conducted to find out the best combination of knife type, unloading system and speed of the feeding system of the mentioned prototype. This paper is good. However, in my opinion following changes are necessary before publication: 

R: thanks for the good evaluation of the ms.

Please clarify more why author specifically choose this particular prototype. If possible, present a comparison with existing other modular chipper. 

1. Please clarify more why author specifically choose this particular prototype. If possible, present a comparison with existing other modular chipper.

R: The prototype has been developed within the European project "Europruning". The peculiarity of the prototype was certainly its versatility and ability to adapt to different farm contexts. This feature of the machine makes it unique and as suggested it has been emphasized in the lines 117-121 of the manuscript (lines refer to the tracked change mode). 

The most important trait of the modular prototype is that it could be modified at farm level in order to adapt it to different harvesting logistic, allowing the unloading of the chipped product on a wagon towed by a tractor at the back or at the side of the prototype, unloading the chopped product on big bags, or in the container that can be installed on top of the prototype. To the best of our knowledge, there are no modular machines in the market or other similar prototypes. However, there are various commercial non-modular shredders on the market that were mentioned in the paragraph 3 of the manuscript. From this point of view, we regret to say that we could have not test other similar prototypes since there is scarce information on the availability of similar prototypes and, at the one time, the project would have not funded a work on other tools. 

2. Please try to improve the English language.

R: Thanks for the suggestion. English has been revised and improved in many lines of the ms.

3. Correct the headings number ?

R: Done

4. Reference citation should be consistent throughout the paper.

R: thanks for the suggestion, the reference were double checked with Mendeley and corrected, where relevant

5. Conclusions are needed to be more specific and concise.

R: thanks for the suggestion. Extensive and redundant part of the conclusions were deleted. Some additional parts were clarified and the fuel consumption per unit yield highlighted at the light of the energy production.

Reviewer #2: Some comments are given as follows:

(1) Lack of significant contributions: The contribution of this paper is not significant. The proposed methodology has limited novelty, and the overall impact is not significant enough;

R: We regret to say that the reviewer evaluation is not fit for either the ms and the journal.

In particular, with regards to the ms topic, the work is not based on the application of an innovative methodology, but on an overall study of the performance and quality of work of a prototype of modular chipper for the collection of pruning residues, using a well-established methodology already applied in countless studies. In addition, the tool studied here is not under a commercial production and the potential impact of the implementation of its modularity (either here or in other tools) may boost the sustainability of the use of similar devices.

In addition, this study fills a gap in the chipper sector by describing the performance of an innovative machine that does not exist on the market: a modular chipper capable of adapting to different harvesting logistics, and with a chipping system based on helical knives, which has been scarcely investigated in the literature. Therefore, the study represents an innovation. Finally, the study applies a complex statistical analysis to compare the data collected in different cropping seasons, on different types of prunings (in term of plant species), various machine settings and different harvesting logistics of the chopped product, reaching a considerable complexity of analysis. This analysis allow to provide a reliable and robust result on the treatments applied.

With regards to the journal, we would like to pinpoint that Plos1 does not take into account the perceived novelty of the work and accepts manuscripts as a publication given that the methodology is correct, as also clearly indicated in the journal website (https://journals.plos.org/plosone/s/journal-information). Nonetheless, additional comments throughout the ms were provided to highlight the novelty of the study. 

(2) Clarity and Presentation: the paper is not well presented, and there are quite some English writing errors/typos. The authors are suggested to check the whole paper carefully to avoid such errors.

R: Thanks for the suggestion. English has been revised and improved and various sentences were made clearer. Please check the ms in the tracked change mode.

---

## [Decision Letter · Decision Letter 1]

4 Nov 2021

PONE-D-21-04650R1Pruning harvesting with modular towed chipper: little effect of the machine setting and configuration on performance despite strong impact on wood chip qualityPLOS ONE

Dear Dr. Saia

Thank you for submitting your manuscript to PLOS ONE. After careful consideration, we feel that it has merit but does not fully meet PLOS ONE’s publication criteria as it currently stands. Therefore, we invite you to submit a revised version of the manuscript that addresses the points raised during the review process.

We look forward to receiving your revised manuscript.

Kind regards,

Tunira Bhadauria, Ph.D.

Academic Editor

PLOS ONE

Journal Requirements:

Reviewers' comments:

Reviewer's Responses to Questions

**Comments to the Author**

1. If the authors have adequately addressed your comments raised in a previous round of review and you feel that this manuscript is now acceptable for publication, you may indicate that here to bypass the “Comments to the Author” section, enter your conflict of interest statement in the “Confidential to Editor” section, and submit your "Accept" recommendation.

Reviewer #1: All comments have been addressed

Reviewer #3: (No Response)

2. Is the manuscript technically sound, and do the data support the conclusions?

Reviewer #1: Yes

Reviewer #3: Yes

3. Has the statistical analysis been performed appropriately and rigorously? 

Reviewer #1: Yes

Reviewer #3: Yes

4. Have the authors made all data underlying the findings in their manuscript fully available?

Reviewer #1: Yes

Reviewer #3: Yes

5. Is the manuscript presented in an intelligible fashion and written in standard English?

Reviewer #1: Yes

Reviewer #3: Yes

6. Review Comments to the Author

Reviewer #1: This manuscript can be accepted. All the questions were answered satisfactorily. I have no further comments.

Reviewer #3: The work done are appreciable. The manuscript is written in good fashion but few things need to be added. The study does not include effect generated by the wear of knives and the influence of the hardness of different woods processed. In my opinion without these results the study is not complete. Another thing, the data and study plan is too small in the MS. If possible, Please include more data in the study

7. PLOS authors have the option to publish the peer review history of their article (what does this mean?). If published, this will include your full peer review and any attached files.

Reviewer #1: No

Reviewer #3: No

---

## [Author Response · Author response to Decision Letter 1]

14 Nov 2021

Rebuttal letter for PONE-D-21-04650R1

Reviewer #1: This manuscript can be accepted. All the questions were answered satisfactorily. I have no further comments.

Authors: thanks for your positive comments, that helped us to improve the ms.

Reviewer #3: The work done are appreciable. The manuscript is written in good fashion but few things need to be added. The study does not include effect generated by the wear of knives and the influence of the hardness of different woods processed. In my opinion without these results the study is not complete. Another thing, the data and study plan is too small in the MS. If possible, Please include more data in the study

Authors: firstly, we would like to thanks for the positive comments, that helped us to improve the ms. Regarding the additional comments received, we would like to pinpoint to both the reviewer and editor that the work indirectly took into account the hardness of different woods processed, since we used woods from different species and growing seasons. In addition, studying the role of the hardness of different woods processed was not a direct aim of the study.

Also, the study of the wear of knives was out of the scope of the present work. Indeed, it may have add interesting information, but the aim of the study was acquiring information on the efficiency of the system in term of time needed, fuel and quality of the product. 

These aspects will likely be included in other works, but not in the present work. 

We’d also like to pinpoint that the completeness of a study is achieved when the methods respond to the aim and not to the number of topics dealt. We thus disagree the opinion of the reviewer that the study is not complete since she/he is proposing us another different study. 

Regarding the data and study plan, all the information to reproduce the work were provided and the complete raw dataset was provided in the supplementary material. We avoided providing other data in the main ms since this would have inflated uselessly the ms. The ms presently have 7 big-sized tables. Nonetheless, all the data produced were included in the supplementary material, including the covariance parameter estimates. We thus fear that the reviewer may have inadvertently missed part of the results provided.

---

## [Editor Report · Decision Letter 2]

13 Dec 2021

Pruning harvesting with modular towed chipper: little effect of the machine setting and configuration on performance despite strong impact on wood chip quality

PONE-D-21-04650R2

Dear Dr. Sala

We’re pleased to inform you that your manuscript has been judged scientifically suitable for publication and will be formally accepted for publication once it meets all outstanding technical requirements.

Kind regards,

Tunira Bhadauria, Ph.D.

Academic Editor

PLOS ONE
---

## [Editor Report · Acceptance letter]

20 Dec 2021

PONE-D-21-04650R2 

Pruning harvesting with modular towed chipper: little effect of the machine setting and configuration on performance despite strong impact on wood chip quality 

Dear Dr. Saia:

I'm pleased to inform you that your manuscript has been deemed suitable for publication in PLOS ONE. Congratulations! Your manuscript is now with our production department. 

Kind regards, 

on behalf of

Dr. Tunira Bhadauria 

Academic Editor

PLOS ONE